# Adaptation of a Commercial NAD^+^ Quantification Kit to Assay the Base-Exchange Activity and Substrate Preferences of SARM1

**DOI:** 10.3390/molecules29040847

**Published:** 2024-02-14

**Authors:** Ilenia Cirilli, Adolfo Amici, Jonathan Gilley, Michael P. Coleman, Giuseppe Orsomando

**Affiliations:** 1Department of Clinical Sciences (DISCO), Section of Biochemistry, Polytechnic University of Marche, Via Ranieri 67, 60131 Ancona, Italy; i.cirilli@staff.univpm.it (I.C.); a.amici@staff.univpm.it (A.A.); 2John van Geest Centre for Brain Repair, Department of Clinical Neurosciences, University of Cambridge, Forvie Site, Robinson Way, Cambridge CB2 0PY, UK; jg792@cam.ac.uk (J.G.); mc469@cam.ac.uk (M.P.C.)

**Keywords:** enzyme assay, SARM1, NAD(P)ase, base exchange, pyridine bases, acetylpyridine, isoquinoline, pyridine dinucleotides, acetylpyridine adenine dinucleotide, NAD/NADH-Glo™ Assay

## Abstract

Here, we report an adapted protocol using the Promega NAD/NADH-Glo™ Assay kit. The assay normally allows quantification of trace amounts of both oxidized and reduced forms of nicotinamide adenine dinucleotide (NAD) by enzymatic cycling, but we now show that the NAD analog 3-acetylpyridine adenine dinucleotide (AcPyrAD) also acts as a substrate for this enzyme-cycling assay. In fact, AcPyrAD generates amplification signals of a larger amplitude than those obtained with NAD. We exploited this finding to devise and validate a novel method for assaying the base-exchange activity of SARM1 in reactions containing NAD and an excess of the free base 3-acetylpyridine (AcPyr), where the product is AcPyrAD. We then used this assay to study competition between AcPyr and other free bases to rank the preference of SARM1 for different base-exchange substrates, identifying isoquinoline as a highly effect substrate that completely outcompetes even AcPyr. This has significant advantages over traditional HPLC methods for assaying SARM1 base exchange as it is rapid, sensitive, cost-effective, and easily scalable. This could represent a useful tool given current interest in the role of SARM1 base exchange in programmed axon death and related human disorders. It may also be applicable to other multifunctional NAD glycohydrolases (EC 3.2.2.6) that possess similar base-exchange activity.

## 1. Introduction

The sterile alpha and TIR motif-containing protein 1 (SARM1) is a protein adapter of Toll-like receptors that plays a key role in programmed axon death [1] and is currently recognized as a potential therapeutic target for a number of neurodegenerative disorders [2,3]. SARM1 is a multifunctional NAD(P) glycohydrolase (NAD(P)ase, EC 3.2.2.6) generating nicotinamide (Nam) and, depending on the activity, ADPR(P) (hydrolysis), cyclic ADPR(P) (cyclization), or potentially NAD or NADP analogs in which other free pyridine bases when they are available are exchanged for Nam (transglycosidation, or “base exchange”) [4,5].

Induction of SARM1 glycohydrolase activity is a central step for executing programmed axon death, and this is triggered by a combination of increasing axonal levels of the key NAD-synthesis intermediate nicotinamide mononucleotide (NMN), or its pyridine analogues, and decreases in NAD [6,7,8,9]. In current models, NMN and NAD compete for binding to a site within the inhibitory armadillo/HEAT motif (ARM) domains of SARM1 octamers: SARM1 activity is normally suppressed while NAD is bound but its displacement by NMN triggers a conformational change in the whole protein that facilitates oligomerization of the catalytic Toll/interleukin-1 receptor (TIR) domains to generate a more active form of the enzyme [10,11,12]. Remarkably, the positive allosteric regulation exerted by NMN on ARM domains applies equally to all of the SARM1 enzymatic activities [5].

While there has been much focus on SARM1 NAD hydrolase activity as the driver of programmed axon death, the role of its base-exchange activity, an equally plausible candidate mechanism, has until recently received less attention [13,14,15]. Uniquely among the mammalian enzymes with transglycosidase activity, base exchange by SARM1 can occur at physiological pH [5], depending on free pyridine bases available to exchange [7,13]. SARM1 is thus expected to catalyze the majority of NAD transglycosidation in vivo. Importantly, calcium mobilization plays a crucial role in axon degeneration and the potent calcium mobilizer, nicotinic acid adenine dinucleotide phosphate, can be formed by SARM1 base exchange from nicotinic acid (Na) and NADP [16]. This represents one way that SARM1 base exchange might contribute to degeneration, although this has not yet been confirmed.

The work described here was initiated after we attempted to use the NAD/NADH-Glo™ Assay to quantify SARM1-mediated consumption of NAD in reactions containing excess 3-acetylpyridine (AcPyr) but observed an increased rather than decreased luminescence signal. Under these conditions, SARM1 almost exclusively catalyzes base exchange to produce 3-acetylpyridine adenine dinucleotide (AcPyrAD) from NAD [5], and so we hypothesized that, given the close similarity between Nam and AcPyr (Figure 1), AcPyrAD might represent an alternative, and better, substrate for the cycling enzymes in the NAD/NADH-Glo™ Assay to account for the enhanced signal in these reactions. This allowed us to devise a new method for assaying the 3-acetylpyridine exchange reaction on NAD catalyzed by SARM1.

## 2. Results

### 2.1. Luminescence Emission in the NAD/NADH-Glo™ Assay and Derived Calculations

The NAD/NADH-Glo™ Assay kit uses two highly selective but undisclosed enzymes, an “NAD Cycling Enzyme” plus a “Reductase”, to reduce proluciferin into luciferin. Light is generated from a “Luciferase” in the presence of ATP that is proportional to the amount of NAD/NADH within a range from 10 nM to 400 nM with measurements typically taken between 30 and 60 min. Notably, temperature may change the rate of light generation due to effects on the enzyme rates.

We obtained experimental RLU measures, at 25 °C, from a standard curve of pure NAD (Appendix A) from 1 pmol to 100 pmol, equivalent to 10 nM to 1000 nM in solution, thus purposefully exceeding the linearity of the assay. As a result, the luminescence profiles generated were parabolic rather than linear, albeit linearity could be seen after an initial short lag (Figure 2A). Most curves then showed a general pattern of plateauing followed by a gradual decline (likely as some of the reagents become limiting). Curve amplitudes increased with increasing NAD concentration, with an earlier plateau and decline phase. From this data set, we calculated RLU/min from best linear fitting of each curve (Figure 2B) and plotted slopes versus corresponding NAD amounts (Figure 2C). Linearity in this case was seen up to 20–40 pmol of NAD (200–400 nM in solution), thus close to the upper threshold limit of the kit. Conversely, when light emission was taken from the initial ramp of each curve and, after a derivative calculation and plotting (Figure 2D, and see Section 4), reported and plotted as RLU/min^2^, we could see improved linearity up to 100 pmol of NAD at least (Figure 2E). As such, this latter calculation was used in all subsequent analyses.

### 2.2. Cycling Amplification of AcPyrAD in the NAD/NADH-Glo™ Assay

Early testing revealed that purified AcPyrAD (see Appendix A) is a substrate for cycling in the NAD/NADH-Glo™ Assay, with the curves acquired being similar to those obtained for NAD (Figure 3A). In fact, AcPyrAD (alone) resulted in more intense light production compared to NAD with signal amplitudes approximately 3.5-fold higher at 25 °C (see Figure 3B and the RLU/min^2^ per pmol of 3.36 for AcPyrAD compared to 0.96 for NAD). On the other hand, AcPyrAD was not cycled in the NADP/NADPH-Glo™ Assay, indicating specificity for the NAD/NADH-Glo™ kit.

### 2.3. Quantification of AcPyrAD When Mixed with NAD

We next assessed the linearity and the amplitude of emitted signals when AcPyrAD and NAD were mixed at various ratios for three different total amounts (1, 2 and 5 pmol). We found a linear relationship for our specific test conditions as light emission (reported as RLU/min^2^) increased proportional to the relative amount of AcPyrAD in the mixture (Figure 3B), with the gradients of the curves being equivalent to each other for the three fixed total amounts of AcPyrAD plus NAD tested (Figure 3C). Re-plotting of the data established that AcPyrAD increased the luminescence signal emitted 248.9% ± 10.7 when displacing the equivalent amount of NAD from the mixtures (Figure 3D). This is also seemingly independent of temperature as similar linearity is obtained when the cycling assay is performed at 35 °C, despite much greater luminescence emitted by both individual NAD and AcPyrAD standards under these conditions (Appendix A). Based on this information we provide details in the Methods of calculations that can be used to determine relative amounts of NAD and AcPyrAD in mixtures. Our quantification as proposed requires that NAD consumed equals the AcPyrAD formed in the solution. Whenever this is not the case, only relative measures on AcPyr base exchange by SARM1 can be achieved (see next paragraph and Section 3).

### 2.4. Using the NAD/NADH-Glo™ Assay to Assay AcPyr Base Exchange on NAD by SARM1

The ability to quantify the relative contributions of each nucleotide to light emission in NAD/NADH-Glo™ Assay in a mixture of NAD and AcPyrAD, as described above, provides a means to assess a change in the NAD/AcPyrAD composition of a reaction mixture. We previously showed that base exchange on NAD (or NADP) is the major activity in vitro when SARM1 is co-incubated with a ~10-fold excess of AcPyr (the hydrolysis and cyclization reactions are both negligible under these conditions) [5]. We thus reasoned that our modified NAD/NADH-Glo™ Assay could be used to monitor SARM1 base exchange with AcPyr (see flowchart in Appendix A). We initially incubated full-length wild-type SARM1 with 50 µM NAD, plus or minus 2 mM AcPyr free base. Reactions were stopped at multiple time intervals and analyzed using the NAD/NADH-Glo™ Assay kit, with each reaction stop corresponding to 2 pmol of NAD in the original reaction, which is within the linear range in Figure 3 for the NAD/NADH-Glo^TM^ Assay. In parallel, to validate the NAD/NADH-Glo^TM^ results, we performed HPLC analysis on the same samples. Comparable product quantifications and final rate values were obtained for both methods (Figure 4), thereby validating our modified NAD/NADH-Glo™ Assay method. Indeed, using a one hour incubation to set the initial velocity, we calculated rates of 9.1 milliU/mg for NAD consumption and of 16 milliU/mg for AcPyrAD accumulation (Figure 4A) using our method that were fully equivalent to recalculations by HPLC based on overall products formed by SARM1 of 8.6 milliU/mg and 13 milliU/mg, respectively (Figure 4B). In absolute terms, an NAD consumption rate that is approximately half of the reported *V*_max_ of ~22 milliU/mg is expected from such a SARM1 preparation given the limiting concentration of NAD used (50 µM is close to the reported *K*_m_ of ~30 µM), as well as the approximately two fold increased rate of AcPyr base exchange [5]. Remarkably, these data were fully replicated at both 25 °C and 35 °C (compare Figure 4A with Appendix A).

AcPyr is also exchanged efficiently by SARM1 on NADP to form AcPyrAD phosphate [5], so we tested this using the NADP/NADPH-Glo™ Assay specific for NADP. We saw a time-dependent decrease in luminescence in parallel assays containing NADP with or without AcPyr consistent with comparable NADP consumption and no amplification arising from the AcPyrAD phosphate product formed by SARM1 (Appendix A).

### 2.5. SARM1 Shows Differing Preferences for Other Free Pyridine Bases When Added in Competition with AcPyr

We next used our method to further investigate SARM1 base exchange. We previously showed that multiple free bases may be exchanged by SARM1 [5], and so we selected the free pyridine bases isoquinoline (isoQ) [17], vacor (1-(4-nitrophenyl)-3-(pyridin-3-ylmethyl)urea) and nicotinic acid (Na) [5,18] and tested them in competition with AcPyr for effects on the transglycosidation of NAD. Preliminary tests showed that the free bases did not interfere with enzymatic cycling in the NAD/NADH-Glo™ Assay.

NAD consumption was first evaluated by the NAD/NADH-Glo™ Assay in SARM1 reaction mixtures containing each free base alone, e.g., 0.5 mM isoQ, or 0.5 mM vacor, or 5 mM Na. Rates resulted all close to ~10 milliU/mg like in NAD alone control (Figure 5, left black lines and right blue lines). HPLC analysis in parallel confirmed this rate value, and also that isoQ exchange represented nearly 100% of the activity, as with AcPyr, while vacor exchange was close to 40%, and Na exchange was less than 3%, as previously reported [5,17]. Thus, there were no changes to the total rate of NAD loss under these conditions when SARM1 activity was switched to varying extents from NAD hydrolysis/cyclization to the synthesis of the corresponding dinucleotides isoquinoline adenine dinucleotide, vacor adenine dinucleotide, or nicotinic acid adenine dinucleotide by base exchange. Importantly, and in contrast to AcPyrAD, these dinucleotides did not act as alternative substrates to NAD in the NAD/NADH-Glo™ Assay as there was no interference with the luminescence emitted by the kit when these dinucleotides were mixed with NAD in the assay.

Next, addition of AcPyr 2 mM to reaction mixtures, as above, allowed us to evaluate the individual contributions of each free base to base exchange by SARM1 (Figure 5, left red lines and right brown lines). We saw the expected rise in light emission with AcPyr alone resulting from AcPyrAD formation (Figure 5A compare with Figure 4A), but this was completely blocked by 0.5 mM isoQ (Figure 5B). In contrast, the same concentration of vacor had no affect (Figure 5C). The relatively more water-soluble Na, at a 10-fold higher concentration of 5 mM, caused only a small, non-significant decrease in AcPyrAD formation (Figure 5D). These data suggest that isoQ is a much better substrate for SARM1, competing very effectively for base exchange with AcPyr, which in turn outcompetes water as a substrate leading to NAD hydrolysis. In contrast, vacor and Na did not compete effectively with AcPyr. Such a difference is likely to reflect different affinities in binding to the catalytic pocket of SARM1. Therefore, on this basis, the order of preferential use of these bases for exchange by SARM1 is: isoQ > AcPyr > vacor/Na.

Remarkably, this assay could be used to test the suitability and preference for other pyridine bases as SARM1 base-exchange substrates, and assess dose dependence via titration, provided that the corresponding dinucleotide, or the base itself, is not interfering with amplification and light emission by the kit.

## 3. Discussion

In this study, we have developed a novel method for assaying the base-exchange activity of SARM1. This method originated from an unexpected observation when utilizing the NAD/NADH-Glo™ Assay, a commercial kit, to investigate the impact of effectors on NAD consumption by SARM1. Notably, our assay is applicable to all NAD(P)ases which may catalyze exchange reactions with AcPyr, such as CD38 and *Aplysia californica* ADP ribosyl cyclase [5]. We have subsequently validated our method through HPLC and demonstrated its potential for future adaptations, such as rapid screening using a microplate assay, which offers greater sensitivity and efficiency compared to HPLC. As such, it requires indeed less than half time with respect to a single HPLC analysis as reported, and it holds the potential to assess simultaneously in one plate the preference of SARM1 for base exchange with multiple other free pyridine bases through their ability to compete with AcPyr. Furthermore, the experimental sample handling is facilitated, and detection sensitivity is high from the kit. Following our protocol’s dilutions (refer to Section 4 and Appendix A), we observed a threshold of around 100 femtomoles, which is at least two orders of magnitude lower than standard HPLC techniques with diode-array detection. Recently, a couple of equally sensitive, fluorescent assays for SARM1 base exchanging have been reported [13,14], some of which were used for screening too, albeit employing synthetic probes as substrates instead of AcPyr, primarily for live-cell imaging of SARM1 activity. The availability of multiple complementary assays for the base exchanges by SARM1 provides a distinct advantage in comprehending the pivotal role played by this multidomain protein and multifunctional, auto-regulated enzyme in programmed axon death.

Crucially, the NAD/NADH-Glo™ Assay kit employes enzymatic cycling to generate a bioluminescence signal corresponding to the NAD/NADH levels in solution. We proved that acceleration in appearance of this signal at early stages of cycling, rather than the rate of appearance, is linear over a broader range of NAD concentrations (see Figure 2). Although the kit manufacturer does not specify the enzymatic mechanism it is based on, this observation could be explained by a luciferin recycling process at each amplification cycle, leading to new light emission in each cycle. Such a luciferin-regenerating system is known to occur in nature working in the presence of d-cysteine [19,20]. Nevertheless, our method remains valid regardless of how luminescence is generated or processed, as long as quantification is conducted within established and verified confidence limits.

Our calculation of mixed quantities of NAD and AcPyrAD stems from the differential efficiency of each as substrates for the enzymes involved in the NAD/NADH-Glo™ Assay kit’s cycling process. This results in 3–4-fold greater amplification with AcPyrAD compared to NAD. Under these circumstances, an increase in luminescence corresponds proportionally to the NAD-to-AcPyrAD conversion in the mixture, at least within verified limits (as depicted in Figure 3). Additionally, as shown in Figure 3D, this signal changes relative to the molar fraction of the two mixed dinucleotides in solution. This conceptually resembles Raoult’s law for ideal binary solutions of liquids, albeit in our case, the total vapor pressure is replaced by the luminescence emitted during cycling.

Our quantification process relies on achieving precise stoichiometry in the conversion of NAD into AcPyrAD. This is not straightforward for multifunctional NAD(P)ases like SARM1, which may catalyze both NAD hydrolysis and cyclization concurrently, thereby generating additional products alongside the exchanged dinucleotide. These additional products, unidentified by the cycling assay, hinder an accurate quantification of how much NAD decline is solely attributable to base exchange, and thus limit our calculation (see Section 4). Furthermore, when NAD(P)ases exchange multiple free bases simultaneously, the complexity of the product mix increases. Ultimately, quantification is feasible only if the parallel hydrolysis and cyclization of NAD are both negligible, as shown in Figure 4.

Despite these challenges, our method still allows for relative measurements of substrate preference. For example, we tailored our SARM1 assay towards a competition between the free AcPyr base and other pyridines. Due to formation of extra dinucleotide products under these conditions, the quantification of AcPyrAD was only relative, as shown in Figure 5, but still useful in establishing SARM1′s preference for each pyridine base tested. Importantly, these bases could be assayed by this means because they have no direct or indirect effects on enzymatic cycling by the kit. This verification needs to be carried out on a case-by-case basis when testing new compounds. However, it is reasonable to assume that most free pyridine bases, including many current drugs belonging to this class, could function similarly. Even in cases where interference may arise, it may be possible to exploit this in the same way as we have with AcPyr/AcPyrAD, and even if the NAD-analog dinucleotide exhibits lower amplification efficiency compared to NAD in the NAD/NADH-Glo™ Assay. Interestingly, all these scenarios could potentially expand the functionality of this assay kit for novel applications on base exchanges by SARM1 or similar NAD(P)ases. In this regard, we also reported preliminary evidence that a related Promega kit, the NADP/NADPH-Glo™ Assay, does not use either AcPyrAD or AcPyrADP as a substrate.

Our finding that SARM1 efficiently exchanges isoQ appears surprising given that isoquinoline derivatives are known as potent inhibitors of SARM1 NAD(P)ase [21]. The isoQ derivative, 5-iodo-isoquinoline, inhibits SARM1 strongly, but only after a SARM1-dependent formation of the corresponding NAD-analog dinucleotide, 5-iodo-isoQ adenine dinucleotide, by base exchange as the effector of that inhibition [15]. Therefore, while the precursor isoQ is an high affinity substrate for SARM1 base exchange, simultaneously reducing both hydrolysis and cyclization reactions without altering the overall rate of NAD consumption [17] (and unpublished data), isoQ adenine dinucleotide does not appear to inhibit SARM1 activity in the same way as its iodinated derivative.

Finally, the predicted displacement of AcPyr by isoQ from the SARM1 catalytic site, and by extension displacement of vacor, may have clinical implications. Both vacor and AcPyr are well-known neurotoxins that activate SARM1 via their mononucleotides formed in vivo by NAMPT, but which are also converted to corresponding dinucleotides by SARM1 base-exchange activity in vivo [5,6,7,18]. The potential effects of such dinucleotides on living cells remain unknown but reducing their levels via isoQ supplementation could provide important further insight into the mechanism of AcPyr and vacor neurotoxicity.

In summary, the method developed here is useful for investigating the mechanism of action of SARM1 in neurodegeneration with potential applications in the development of therapies for SARM1-related neurological disorders. It may also facilitate future studies on NAD transglycosidation in cells and the NAD analogs generated through base-exchange reactions. It has a particular use, as we have shown, in testing the preference of SARM1 for different base-exchange substrates that compete with AcPyr to varying degrees. The pressing need for assay methods tailored to pyridine bases exchanges underscores the significance of this tool, which may undergo further refinements and enhancements in the future.

## 4. Methods

Reagents. The NAD/NADH-Glo™ Assay kit (#G9071, LOT: 0000469987) and the NADP/NADPH-Glo™ Assay kit (#G9081, LOT: 0000472344) were purchased from Promega (Madison, WI, USA). Vacor (Pyrinuron N-13738) was from Greyhound Chromatography (Birkenhead, UK). Other chemicals were from Merck (Lowe, NJ, USA) (at the highest purity) and used without additional treatment, except for NAD (Merck N1511) and AcPyrAD (Merck A5251) that were both purified by FPLC ion exchange chromatography (IEC). Briefly, a TSK DEAE column (Tosoh, 4.6 × 250 mm) was equilibrated at 1 mL/min (see buffers in Appendix A), then loaded with each dinucleotide above dissolved in water (0.3–2 micromoles per run) followed by a salt gradient elution. The peaks corresponding to NAD or AcPyrAD were pooled from multiple runs, quantified by UV absorption (ɛ260 nm of 18 mM^−1^ cm^−1^ or 16.4 mM^−1^ cm^−1^), frozen, and lyophilized. Once resuspended, they were confirmed as 100% pure by C18-HPLC analysis (Appendix A). They were stored at −80 °C until use.

Enzyme preparation. Full-length human SARM1 (wild type) was expressed in HEK cells, purified on magnetic beads, and quantified by immunoblotting as described [5]. Beads suspensions in PBS buffer plus BSA 1 mg/mL were stored at −80 °C in aliquots and used without additional freeze/thawing.

Bioluminescence assay conditions. Cycling reactions were performed by the NAD/NADH-Glo™ Assay or by the NADP/NADPH-Glo™ Assay in 96-well plates using 100 µL per well, i.e., 50 µL of sample and 50 µL of the kit’s reagents as indicated by the manufacturer. Luminescence readings were recorded using a Sinergy HT microplate at sensitivity 100, under shaking intensity of 4 (2 s before every reading), and using a top probe vertical offset of 1 mm. No temperature control was set (so 25 °C) except when specified. Luminescence emitted (RLU) was monitored and recorded typically for 2 h, then the light outputs were processed by Excel to calculate first derivatives (delta RLU increments per min), usually within the initial ramp of each curve (2–25 min of records). The RLU increments were plotted versus time and linearized. Slopes from best linear segments obtained (RLU/min^2^), for long as possible (>5 min at least), were plotted against pmol of NAD or AcPyrAD or the two of them mixed at different ratios. In some cases, luminescence was referred to as relative percentage of the initial NAD concentration arbitrarily fixed to 100%.

Enzyme assay and rate calculation. Control assay mixtures for NADase were carried out at 25 °C in 50 mM Hepes/NaOH buffer, pH 7.5, in the presence of 7.5–12 ng/µL full length, wild-type SARM1 and 50 µM NAD (or NADP). Assay mixtures for base exchange were set in parallel by further adding 2 mM AcPyr and/or 0.5 mM to 5 mM of other free pyridine bases as indicated in the legend of Figure 5. Reactions were stopped at 1 h intervals (10 µL each) and mixed with ice-cold 1.2 M HClO_4_ (+5 µL each to reach 0.4 M final), then centrifuged, diluted to 1:100 in water, and frozen. Subsequent analyses were done by cycling as above or, when needed to validate the proposed assay procedure, by HPLC. So, 4 µL aliquots of each diluted reaction stop (2 pmol of original NAD(P) in the mixture) were thawed and cycled, or 400 µL aliquots (200 pmol original NAD) were loaded for C18-HPLC. HPLC data were calculated as described previously [5] while calculations from cycling were as follows. Pure standards of NAD and AcPyrAD were cycled together with reaction stops (thus under same temperature conditions) and used for converting their luminescence signals. So, the slope of the NAD standard curve was used to quantify NAD amounts in control mixtures from Equation (1), while AcPyrAD amounts from base-exchange reactions were calculated from the modified Equation (2).
(1)$$x=\frac{y(t)}{A}$$
(2)$$x=\frac{y(t)-y\left(t0\right)}{\left(B-A\right)}$$
where *y*(*t*) and *y*(*t*0) represent experimental RLU/min^2^ values at each time stop (*t*) and at start of the time-course analysis (*t*0); “A” represents the RLU/min^2^ per pmol of the NAD standard (e.g., 0.96 at 25 °C, see Figure 3B); “B” represents the same value calculated for the standard AcPyrAD (e.g., 3.36 at 25 °C, see Figure 3B). As a result, “x” from Equation (1) corresponds to pmol of NAD alone in control mixtures while, from Equation (2), it corresponds to pmol of AcPyrAD formed via base exchanging, thus matching the NAD-to-AcPyrAD conversion occurred in the mixture. Indeed, the *t*0 stop from AcPyr exchanging reactions is used twice, first to determine the initial NAD concentration in the mixture (from Equation (1)), then to calculate any amount of AcPyrAD formed at subsequent time stops (from Equation (2)). Worth mentioning, Equation (2) above allows correct quantitation only when no other products than AcPyrAD are formed by the NAD(P)ase in parallel, and this depends essentially on assay conditions, substrates used, etc. Furthermore, in AcPyr exchanging reaction mixtures, once AcPyrAD formed amounts are set from Equation (2), corresponding NAD amounts are calculated by difference between initial NAD and formed AcPyrAD. All enzyme rates are referred to as the Unit (U) of activity that corresponds to 1 µmol/min of substrate consumed or of all products formed by SARM1 at 25 °C.

An example of this procedure is summarized as a flowchart diagram in Appendix A.

## Figures and Tables

**Figure 1 molecules-29-00847-f001:**
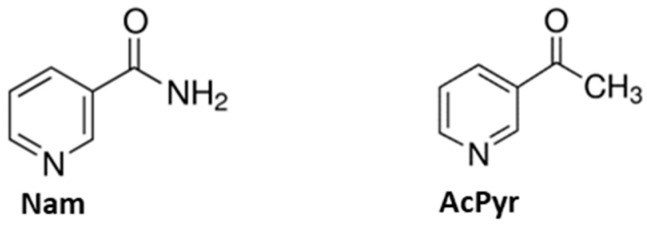
Structural similarity between nicotinamide (Nam) and 3-acetylpyridine (AcPyr).

**Figure 2 molecules-29-00847-f002:**
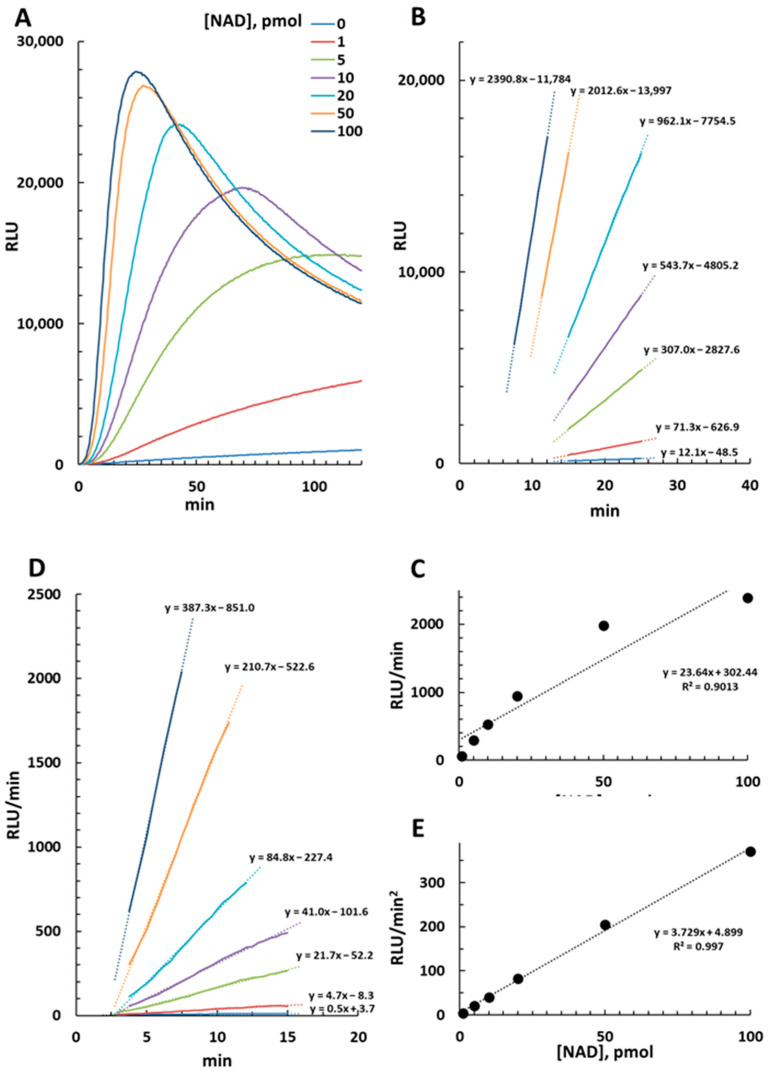
NAD/NADH-Glo™ Assay at 25 °C of NAD alone. Luminescence emission (RLU) from 0 to 100 pmol of NAD was monitored for 2 h and recorded (**A**), then data outputs were taken under linearity ranges and confidence limits as shown to calculate either RLU/min (**B**) or RLU/min^2^ (**D**) values. Slopes for each calculation were finally plotted versus corresponding NAD amounts and analyzed for linearity (**C**,**E**). Data in (**A**,**B**,**D**) are color coded and show R^2^ ≥ 0.99 after linearization.

**Figure 3 molecules-29-00847-f003:**
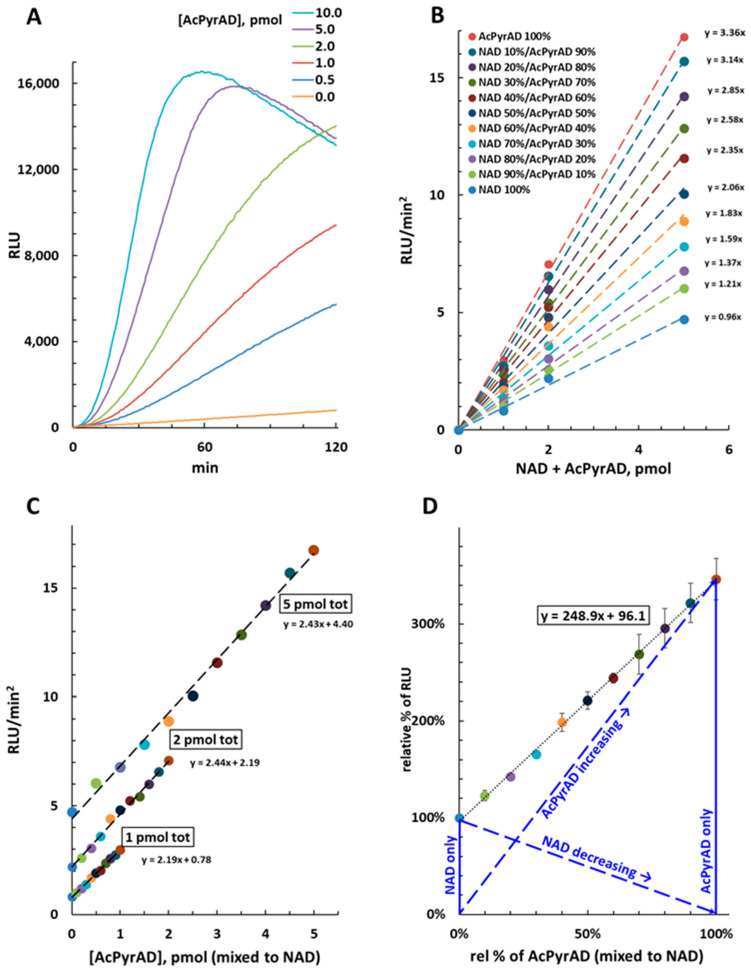
NAD/NADH-Glo™ Assay at 25 °C of AcPyrAD alone or mixed to NAD. (**A**) Luminescence emission (RLU) from 0 to 10 pmol of AcPyrAD. (**B**) Recalculated luminescence emission values (RLU/min^2^, obtained as in Figure 2) from 1, 2 and 5 pmol of NAD or AcPyrAD or both mixed at the different ratios as indicated. (**C**) Replotting of panel B data versus absolute amounts of AcPyrAD in each set. This shows linearity of the luminescence emitted and invariability of the slopes obtained by amplifying different total amounts of NAD plus AcPyrAD (thus 1, 2, or 5 pmol). (**D**) Replotting of panel B data after conversion into relative percentages of initial NAD values (thus 1, 2, or 5 pmol, each fixed to 100%) versus relative percentages of AcPyrAD. Linearization highlights a value of approximately 250% that corresponds to the relative increase in the luminescence signal emitted by cycling reactions at 25 °C for any amount of AcPyrAD ≤ 5 pmol when directly replacing NAD in the mixture. Blue lines graphically illustrate the parameters and the rational of the method employed for rate calculation (fully detailed in Section 3 and Section 4). Data in (**B**–**D**) are color coded and show R^2^ values ≥ 0.99 in all cases after linearization.

**Figure 4 molecules-29-00847-f004:**
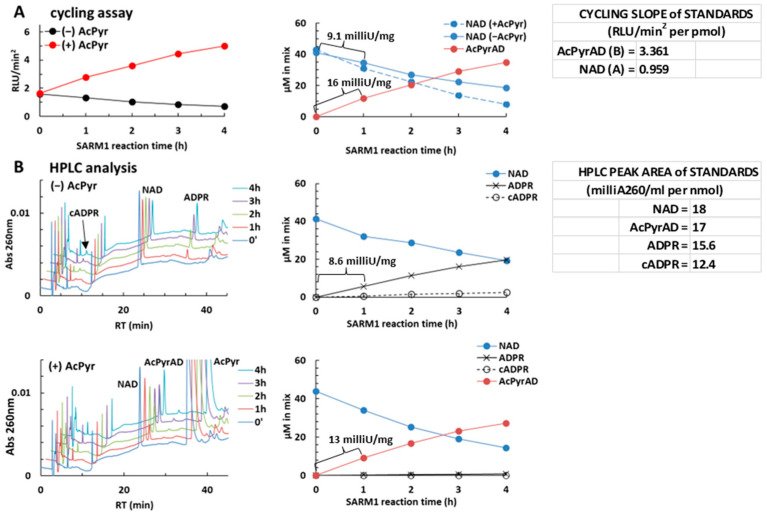
A typical reaction of AcPyr base exchange on NAD catalyzed by SARM1 evaluated by NAD/NADH-Glo™ Assay at 25 °C and C18-HPLC in comparison. Reactions containing recombinant SARM1 and 50 µM NAD with or without 2 mM AcPyr were set, incubated, and processed as reported in Methods and Appendix A. Thereafter, 4 µL aliquots of reaction stops, diluted 1:100, were analyzed by cycling at 25 °C at the timepoints indicated (**A**), together with parallel analysis by C18-HPLC of 400 µL aliquots of the same diluted samples (**B**). Luminescence (RLU) recorded after cycling was processed as in Figure 2 to calculate RLU/min^2^ values and then the levels of NAD consumed in the control ((**A**) right, continuous blue line) or of AcPyrAD formed in the mix with NAD plus AcPyr ((**A**) right, brown line). Calculations were made using formulas in Methods and the indicated standard slopes (**A**,**B**). NAD amounts in the mix with NAD plus AcPyr ((**A**) right, dotted blue line) were calculated by difference (see Section 4) given the whole SARM1 rate here represented by AcPyr base exchange only under these conditions. Alternatively, for panels in (**B**), substrates and products were separated by C18-HPLC and quantified from their peak areas using the standard values as indicated. This led to independent calculation of final concentrations in both original mixtures ((**B**) right panels) to make a comparison. The rates for full-length wild-type SARM1 were calculated from NAD consumed or the indicated products formed within the first one hour of incubation.

**Figure 5 molecules-29-00847-f005:**
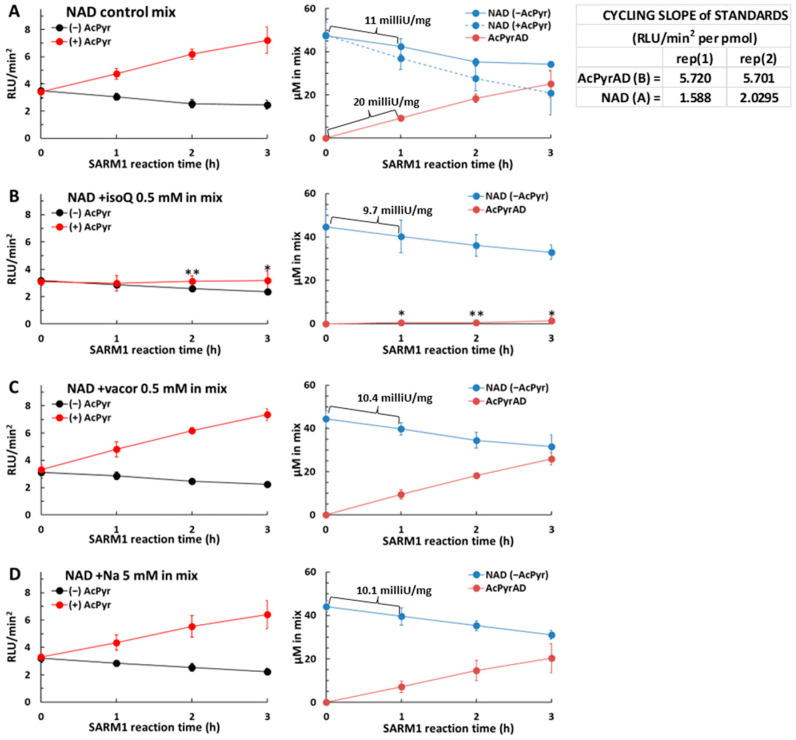
AcPyr base exchange on NAD by SARM1 in the presence of isoquinoline, vacor, and nicotinic acid. Reaction mixtures containing recombinant SARM1 and 50 µM of NAD alone (**A**) or plus 0.5mM isoQ (**B**) or plus 0.5 mM vacor (**C**) or plus 5 mM Na (**D**) were set. Each mixture was duplicated with and without 2 mM AcPyr. Reaction stops at the timepoints indicated were cycled at ~27 °C and the RLU plots processed to determine RLU/min^2^ values. Then, NAD amounts in (−) AcPyr mixtures and AcPyrAD amounts in (+) AcPyr mixtures were calculated, respectively, from equations 1 and 2 in Methods, using the indicated standard slopes A and B. Note that NAD levels in (+) AcPyr mixtures are shown in A (dotted blue line) but missing in other panels where extra formed products are present and not measured by this assay; consequently, AcPyrAD levels in (**B**–**D**) are only relative measures (see Section 4). Data are from two independent experiments (mean ± SD; one-way ANOVA with Dunnett’s test, * *p* < 0.05, ** *p* < 0.01). Rates of full-length wild-type SARM1 were calculated from NAD consumed, or AcPyrAD formed in (**A**), within selected time windows of 1 h as shown.

## Data Availability

Data linearization and R-squared analysis were performed using the Microsoft Excel 365 Apps for Enterprise v16.0.17231.20194. Statistical analysis was carried out by one-way ANOVA with Dunnett’s multiple comparison test using the R free software at https://www.R-project.org/, accessed on 27 October 2023. Student’s *t* test *p* values ≤ 0.05 were considered significant as indicated on figures.

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
