# Peer review of "Adaptation of a Commercial NAD+ Quantification Kit to Assay the Base-Exchange Activity and Substrate Preferences of SARM1"

_molecules, 2024, doi:10.3390/molecules29040847_

Round 1

Reviewer 1 Report

Comments and Suggestions for Authors

The paper written by Orsomando et al. describes the development of a new rapid, sensitive and cost-effective method for assaying the base exchange activity of SARM1 by using the Promega NAD/NADH-Glo Assay kit. This method originates from the observation that the NAD analog 3-acetylpyridine adenine dinucleotide also acts as a substrate for this assay and generates signals of larger amplitude than those obtained with NAD.

The method is useful to investigate the mechanism of action of SARM1 and may be also applicable to other multifunctional NAD glycohydrolases which catalyse exchange reactions with AcPyr. It is a very interesting study and relevant results are reported, considering the urgent need for assay methods tailored to pyridine bases exchanges.

The paper is well written, and the method is well described and characterized. Publication in Molecules is strongly recommended.

Minor comments:

-Please define all the abbreviations on first appearance in the manuscript text.

-Please define (A), (B), (C), (D) and (E) in Figure 2 footnote.

-Please check font and size of characters of Figure 4 and 5 footnotes.

-Please check the reference list (e.g. in references 6, 16 and 18 page numbers are missing).

Author Response

Response to Reviewer 1 Comments

1. Summary

Thank you very much for taking the time to review this manuscript. Please find the detailed responses below and the corresponding revisions/corrections highlighted in red in the re-submitted files.

2. Questions for General Evaluation

Reviewer’s Evaluation

Response and Revisions

Does the introduction provide sufficient background and include all relevant references?

Yes

none

Are all the cited references relevant to the research?

Yes

none

Is the research design appropriate?

Yes

none

Are the methods adequately described?

Yes

none

Are the results clearly presented?

Yes

none

Are the conclusions supported by the results?

Yes

none

3. Point-by-point response to Comments and Suggestions for Authors

Comments 1: The paper written by Orsomando et al. describes the development of a new rapid, sensitive and cost-effective method for assaying the base exchange activity of SARM1 by using the Promega NAD/NADH-Glo™ Assay kit. This method originates from the observation that the NAD analog 3-acetylpyridine adenine dinucleotide also acts as a substrate for this assay and generates signals of larger amplitude than those obtained with NAD. The method is useful to investigate the mechanism of action of SARM1 and may be also applicable to other multifunctional NAD glycohydrolases which catalyse exchange reactions with AcPyr. It is a very interesting study and relevant results are reported, considering the urgent need for assay methods tailored to pyridine bases exchanges.

The paper is well written, and the method is well described and characterized.

Publication in Molecules is strongly recommended.

Response 1: We thank the reviewer for this positive comment.

Minor Comments:

Please define all the abbreviations on first appearance in the manuscript text.

Done

Please define (A), (B), (C), (D) and (E) in Figure 2 footnote.

Figure 2 footnote was previously within the text and has been now re-formatted

Please check font and size of characters of Figure 4 and 5 footnotes.

They both have been corrected as suggested

Please check the reference list (e.g. in references 6, 16 and 18 page numbers are missing).

The list has been checked and corrected as suggested

Reviewer 2 Report

Comments and Suggestions for Authors

The manuscript authored by Ilenia Cirilli et al presents an assay for measuring SARM1's activity and substrate preference through a combination of the base-exchange reaction between AcPyr and NAD, along with a commercial NAD quantification kit. The clarity of the manuscript's writing is commendable, yet the method described appears to be somewhat indirect and complicated, potentially limiting its widespread adoption in the field.

Major Concern:

In the abstract, the authors assert that their method offers "significant advantages over traditional methods for assaying SARM1 base exchange as it is rapid, sensitive, cost-effective, and easily scalable." To support this claim, a comparison between their method and traditional approaches should be provided, specifically addressing aspects such as speed, sensitivity, cost-effectiveness, and scalability. Furthermore, the paper appears to overlook the PC-probes reported by Li et al. (2022) and Huang et al. (2023), which originally introduced SARM1 probes based on the base-exchange reaction. It is crucial for the authors to address and compare their method with these relevant findings.

Minor Points:

Consider including information labeled as "data not shown" in the supplementary figures to enhance the completeness and transparency of the manuscript. This addition can provide readers with a more comprehensive understanding of the research conducted.

Author Response

Response to Reviewer 2 Comments

1. Summary

Thank you very much for taking the time to review this manuscript. Please find the detailed responses below and the corresponding revisions/corrections highlighted in red in the re-submitted files.

2. Questions for General Evaluation

Reviewer’s Evaluation

Response and Revisions

Does the introduction provide sufficient background and include all relevant references?

Must be improved

not done, see below

Are all the cited references relevant to the research?

Yes

none

Is the research design appropriate?

Can be improved

done, see below

Are the methods adequately described?

Yes

none

Are the results clearly presented?

Can be improved

done, see below

Are the conclusions supported by the results?

Must be improved

done, see below

3. Point-by-point response to Comments and Suggestions for Authors

Comments 1: The manuscript authored by Ilenia Cirilli et al presents an assay for measuring SARM1's activity and substrate preference through a combination of the base-exchange reaction between AcPyr and NAD, along with a commercial NAD quantification kit. The clarity of the manuscript's writing is commendable, yet the method described appears to be somewhat indirect and complicated, potentially limiting its widespread adoption in the field.

Major Concern:

In the abstract, the authors assert that their method offers "significant advantages over traditional methods for assaying SARM1 base exchange as it is rapid, sensitive, cost-effective, and easily scalable." To support this claim, a comparison between their method and traditional approaches should be provided, specifically addressing aspects such as speed, sensitivity, cost-effectiveness, and scalability. Furthermore, the paper appears to overlook the PC-probes reported by Li et al. (2022) and Huang et al. (2023), which originally introduced SARM1 probes based on the base-exchange reaction. It is crucial for the authors to address and compare their method with these relevant findings.

Response 1: We thank the reviewer and have modified according to his insightful comment the Results section (pag 9, lines 260-261 & 263-266) where we have added additional details about our methodology in comparison to the standard HPLC method with diode-array detection used for validation. Responding to the referee’s recommendations, we have included specific information on relative speed, sensitivity, the ease of sample handling, etc. associated with our procedure.

Additionally, as per the reviewer’s advice, we have further referenced Li et al. (2021) and Huang et al. (2023) in the Results section (pag 10, lines 267-271). Both papers were already cited in Introduction (so no changes on it). On this regard, it is important to highlight that our method is significant independently of existing equivalent techniques. Currently, investigations into SARM1 base exchange reactions in neurodegeneration are ongoing, and the specific base exchange product(s) involved remain(s) unknown. Hence, the comparison between assays becomes crucial when employing similar substrates or techniques. Our approach, focusing on AcPyr exchanges through cycling after reaction stopping, differs from the described methodologies where permeable pyridine-modified probes are directly converted into fluorescent products. Moreover, the described methodologies are primarily designed for live cell imaging of SARM1 activity, which is not achievable with our discontinuous assay method.

Regarding instead the perceived complexity and potential limitation for widespread adoption, we have included a supplementary flowchart scheme (now Fig S4) that provides a stepwise description of the entire procedure for assay, data analysis, and rate calculation. We believe this visual aid will enhance the clarity and facilitate understanding for general readers making our method more accessible to researchers.

Minor Points: Consider including information labeled as "data not shown" in the supplementary figures to enhance the completeness and transparency of the manuscript. This addition can provide readers with a more comprehensive understanding of the research conducted.

We deliberated on including data labeled as "not shown"; however, these data primarily consist of negative evidence, such as the absence of cycling observed with AcPyrAD by the NADP-glo kit (page 4, line 112) or the lack of interference on NAD cycling by free pyridine bases (page 7, line 204). Alternatively, some pertain to HPLC data (page 7, line 208) conducted in parallel to validate Figure 5, which has already been presented in a prior publication (referenced as ref 5). In essence, we believe that these data do not provide substantial additional information to the reader. Instead, in response to the referee's suggestion, we have introduced a new flowchart scheme (refer to Fig S4) with the aim of offering a more comprehensive understanding of our method protocol.

Round 2

Reviewer 2 Report

Comments and Suggestions for Authors

“This has significant advantages over traditional methods for assaying SARM1 base exchange as it is rapid, sensitive, cost-effective, and easily scalable.” The authors should specify “traditional methods” as HPLC methods.

Again, this method is too complicated and has limited value compared to the PC6 assay (Li et al, 2021). The author should state the limitations more clearly.
